# The Thermal Tolerance of Springtails in a Tropical Cave, with the Description of a New *Coecobrya* Species (Collembola: Entomobryidae) from Thailand

**DOI:** 10.3390/insects16010080

**Published:** 2025-01-15

**Authors:** Nongnapat Manee, Louis Deharveng, Cyrille A. D’Haese, Areeruk Nilsai, Satoshi Shimano, Sopark Jantarit

**Affiliations:** 1Division of Biological Science, Faculty of Science, Prince of Songkla University, Hat Yai, Songkhla 90110, Thailand; nongnapat.manee@gmail.com; 2Institut de Systématique, Evolution, Biodiversité (ISYEB)—UMR 7205 CNRS, Muséum National d’Histoire Naturelle, Sorbonne Université, 45 rue Buffon, 75005 Paris, France; dehar.louis@wanadoo.fr; 3Mécanisme Adaptatifs & Evolution (MECADEV)—UMR 7179 CNRS, Muséum National d’Histoire Naturelle, Sorbonne Université, 45 rue Buffon, 75005 Paris, France; dhaese@mnhn.fr; 4Excellence Center for Biodiversity of Peninsular Thailand, Faculty of Science, Prince of Songkla University, Hat Yai, Songkhla 90110, Thailand; 5Faculty of Science and Digital Innovation, Thaksin University, 222, Papayom District, Phatthalung 93210, Thailand; areeruk.n@tsu.ac.th; 6Science Research Center, Hosei University, Fujimi, Chiyoda-ku, Tokyo 102-8160, Japan; sim@hosei.ac.jp

**Keywords:** breeding experiment, cave species, global warming, life history traits, thermal tolerance, taxonomy

## Abstract

A new species of Collembola, *Coecobrya microphthalma* **sp. nov.**, is described from a cave in Saraburi province, central Thailand. This species is the second *boneti*-group member found in the country. It closely resembles *C. chompon* Nilsai, Lima & Jantarit, 2022 but differs in having orange body dots and distinct morphological traits, such as the number of sublobal hairs and mac on various body segments. A comparison of all *boneti*-group species globally and a key to their identification are provided. *Coecobrya microphthalma* **sp. nov.** was cultured in the laboratory, and its thermal tolerance was tested at seven different temperatures (27 °C as control, 30, 32, 33, 34, 35, and 36 °C). The results showed that it cannot survive above 32 °C after 7 and 14 days of exposure. At 27, 30, and 32 °C, the species remained alive and produced eggs, though egg-laying duration and number of days decreased with higher temperatures. At 32 °C, the F1 generation survived and molted to adulthood, but no further offspring were produced. Development from egg to adult required six molts, with development rates increasing with higher temperatures. This study is the first attempt to examine how temperature affects the population dynamics, reproductive capacity, and life history of a subterranean tropical Collembola.

## 1. Introduction

The genus *Coecobrya* Yosii, 1956, is one of the largest subterranean genera of springtails and is globally distributed, particularly in East and Southeast Asia [1,2,3,4]. They are characterized by the absence of body scales, labral papillae and dental spines, polymacrochaetotic dorsal chaetotaxy, and a falcate mucro with a basal spine [5,6]. The genus is typically separated into two species groups: the *tenebricosa-* and *boneti*-groups [7], based on the absence of eyes in the former and the presence of eyes (1+1 to 3+3) in the latter. To date, 70 valid species of *Coecobrya* have been recorded worldwide, of which 61 species (87%) have been assigned to the *tenebricosa*-group, and only nine species (13%) belong to the *boneti*-group [4,8,9,10].

In Thailand, the genus *Coecobrya* is, to date, the most diversified and dominant genus found in caves [3,11,12]. A total of 23 known species are recognized in the country [3,4,10,11]. In previous studies, all known *Coecobrya* species of Thailand, which were collected from both subterranean and epigean environments, belonged to the *tenebricosa* group. The cave species *C. chompon* Nilsai, Lima & Jantarit 2022 [4], recently described, represents the first occurrence of the *boneti*-group in Thailand [4], suggesting that this group might be widely distributed in Thai caves, given the uneven zoological investigation in the country. In this study, we describe a second new species of the *boneti*-group found in a cave habitat from the Central plain of Thailand and provide an updated key to the world species of *Coecobrya* of the *boneti*-group.

After direct and severe human disturbance, global climate change is widely recognized as the major threat that may drive biodiversity loss and local extinctions worldwide in the near future [13], influencing the phenology and geographical distribution of a wide range of taxa. Additionally, a major challenge in the fields of ecology, biogeography, and evolution is understanding and predicting how species respond to environmental changes, particularly within this context [14]. The study of thermal tolerance is thus the first step to understanding species’ vulnerability to ongoing climate warming. All organisms are sensitive at various degrees to fluctuations in thermal conditions, as ambient temperature directly impacts their physiological processes and behavior. Rapid climate change, particularly in tropical regions, is heightening the risk of extinction. There remains a dearth of studies concerning the effects of temperature on springtails within cave ecosystems [15], though the work of Thibaud (1970), often overlooked in the literature, has brought a wealth of information for temperate cave Poduromorpha [16].

According to the model CMIP6 [17], air temperatures are expected to increase 2–3 °C by 2050, with a rise of 3–5 °C by 2100. The vulnerability of a species to global warming is determined by its ability to (i) sustain its current population, which is inherently linked to the breadth of its ecological niche, and (ii) shift its geographical range to suitable habitats, a process that depends not only on the quantity and distribution of future available habitats but also on the species’ dispersal capacity [18,19]. The Climatic Variability Hypothesis (CVH) also suggests that climatic variability (rather than a stable climate) plays a key role in shaping the evolution of species, particularly in relation to biodiversity, adaptation strategies and resilience. The hypothesis highlights how short-term or long-term climate changes affect species’ adaptive capacities and environmental stability [20] and further points out that poikilothermic animals in variable temperate climates possess broad thermal tolerance ranges, while those in stable tropical climates have narrow thermal tolerance ranges [21,22]. Hence, accurate predictions of species’ responses are essential for developing effective management strategies in the context of global warming. Unfortunately, quantitative evidence for the impact of climate change on cave environments is very limited, especially in tropical regions where such evidence is nonexistent. Caves serve as ideal natural laboratories for studying climate change due to their climatic stability. Caves have had almost constant climatic factors over the years [23,24]. Subterranean ecosystems are also home to various kinds of organisms, most of which are endemic or unknown to science. Among them, obligate cave-dwellers (i.e., troglobionts and stygobionts) are, physiologically, among the most fragile and vulnerable species on earth [25]. They have evolved to live in dark and nutrient-limited environments under a narrow set of environmental conditions. Many of them have small spatial distribution, and the destruction of their habitats caused by human activities like mining may lead some species to the brink of extinction [26]. The impact of less drastic alterations on the subterranean ecosystem is less predictable due to the diversity of disturbance and species responses ([27] versus [28]). In this line, global warming, which affects all habitats on earth, tends to render subterranean habitats increasingly less favorable for the adapted cave fauna they host, inducing at some point faunistic shifts and even extinctions. These impacts are, however, poorly understood because organism response to temperature changes remains undocumented in most cave-obligate species.

Regarding Collembola, which are usually the dominant arthropods in caves worldwide, the only substantial data are those of Thibaud (1970) [16]. In this work, he brought detailed information on the thermal tolerance and reproductive biology of several cave Poduromorpoha from Europe, bred at different temperatures. Such data are lacking for the order Entomobryomorpha, the most diverse and abundant Collembola underground. The second part of our paper is a contribution to fill this gap, where we present the result of laboratory research on the effect of temperature on the reproduction and life history of *Coecobrya microphthalma* **sp. nov.**, the new cave species described in the first part of this paper.

## 2. Materials and Methods

### 2.1. Taxonomic Study

#### 2.1.1. Sampling and Morphological Identification

The Collembola specimens were collected with aspirators. Some were preserved in 95% ethanol after extraction; others were kept alive for laboratory experiments (see below). Specimens in ethanol were cleared using Nesbitt’s solution and subsequently mounted on glass slides with Marc André II solution. External morphological characters were examined using an Olympus BX51 microscope, Olympus Corporation, Tokyo, Japan, with phase contrast. Photographs were taken with a Canon 5D digital camera using a Canon MP-E 65mm Macro Photo Lens and Canon Extender EF 2.0× III (Canon, Tokyo, Japan) and a Stack-Shot Macrorail (Cognisys Inc., Jackson, MI, USA). Photographs were then combined in Helicon Focus 6.8.0 (Helicon Soft Ltd., Kyiv, Ukraine). For Scanning Electron Microscopy (SEM) photographs, the specimens were cleaned before being dehydrated in ethanol (99%), then processed by being critically dried with CO_2_ (CPD) in a tousimis Autosamdri^®^-931 dryer, Tousimis, Rockville, Maryland, USA. Finally, they were sputter-coated with gold using a Denton Vacuum Desk V and examined using an Apreo SEM/FEI from the Scientific Equipment Center, Prince of Songkla University (Songkhla, Thailand). All photographs were subsequently processed using Adobe Photoshop CC (Adobe Systems Inc., version 26.1.0, San Jose, CA, USA).

#### 2.1.2. Terminology

The pattern of dorsal head and labial basal chaetotaxy is primarily based on the works of Jordana & Baquero (2005) [29], Soto-Adames et al. (2008) [30], Zhang & Pan (2020) [31] and Bellini et al. (2022) [32]. The labial palp papillae and guard-chaetae followed Fjellberg (1999) [33], and the clypeal chaetotaxy followed Zhang et al. (2016) [34]. The antennae III organ and postlabial chaetotaxy were described following Chen & Christiansen (1993) [35] and Jantarit et al. (2019) [3]. The number of dorsal macrochaetae in the descriptions from Th. II to Abd. V is provided by half-tergite. They were labeled following the nomenclature of Szeptycki (1979) [36] and Zhang et al. (2019) [37]. The S-chaetae system used was adapted from Zhang & Deharveng (2015) [38].

Abbreviations used in the description:
Ant.antennal segmentAbd.abdominal segmentGr.cephalic group of chaetaemacmacrochaeta(e)mesmesochaeta(e)micmicrochaeta(e)msS-microchaeta(e)/microsensillum(a)psppseudopore(s)sordinary S-chaeta(e)/senstitatibiotarsustrictrichobothrium(ia)Th.thoracic segmenta.s.l.above sea levelRHrelative humidityhhourLT_50_median Lethal TimeSEMScanning Electron Microscope



### 2.2. Culture and Experimental Design

#### 2.2.1. Colony Maintenance

The living individuals of *Coecobrya microphthalma* **sp. nov.** (the newly described species) were collected by entomological aspirators in situ using visual searching. Once collected, specimens were kept in plastic vials (30 mL) containing soil and organic matter from the original habitats to maintain cool and humid conditions. Additional soil and organic matter were also collected from the cave. The Collembola were then transported immediately to the laboratory at the Biology Department, Faculty of Science, Prince of Songkla University. In the laboratory, specimens were placed inside plastic containers (70 mL), and soil from the original habitats was added to the plastic containers. *Coecobrya microphthalma* **sp. nov.** was reared in the laboratory under controlled temperature and humidity, which were the same as in the original habitats (26–27 °C and 80–90% humidity) to minimize stress. Humidity was controlled by adding tissue paper inside the plastic containers, that was wetted daily with de-ionized water to maintain humid and cool conditions. The plastic containers were sealed with plastic film and punctured with small holes to allow for aeration. Collembola were reared in mass in a dark condition, like in a cave environment. Individuals were fed *ad libitum* (two times a week) with yeast to avoid any potential stress from starvation. Stock populations were reared separately in plastic boxes with finely ground sand mixed with charcoal powder (9:1) as substrate. The animal ethics were approved in accordance with the national guidelines stipulated by the Institutional Animal Care and Use Committee of the Prince of Songkla University (approval no. U106352-2560).

#### 2.2.2. Experimental Design

To study the impact of temperature upon subterranean Collembola, *Coecobrya microphthalma* **sp. nov.**, two complementary series of experiments were conducted: one for thermal tolerance and one for life history. Seven different temperatures were used for the thermal tolerance study; they are 27 °C (as control), 30, 32, 33, 34, 35, and 36 °C and 30 Collembola individuals were used for each experiment. The detailed study is described below.

(1) Thermal tolerance experiments

*Coecobrya microphthalma* **sp. nov.** were taken from the stock population and placed in plastic containers, in darkness and constant high RH (80–90%) conditions. Food was also provided *ad libitum*. To understand the impact of temperature change upon subterranean Collembola, the F1 and F2 generations were tested in the experiments to minimize the parental and potential carryover effects from the environment of origin. Also, using the F1 and F2 generation would reduce the cellular damage (that can occur during the specimen collection of F0), as the result of cellular damage can be linked to exposure to extreme temperatures. To estimate the species heat tolerance survival was assessed at various temperatures over a seven-day period of long-term exposure, given that complete responses typically occur within a shorter time frame in terrestrial arthropods [39]. The Collembola were then continued to be reared in the laboratory for seven more days (14 days in total) to confirm their survival and ability to maintain the population.

According to the expectation of the IPCC (2023) [16], average global temperatures will continue to rise, reaching 2–3 °C by 2050 and 3–5 °C by 2100. Therefore, seven different temperature treatments were set up to test the thermal tolerance of the subterranean Collembola to these treatments (Figure 1). The temperatures used in these experiments were as follows: a normal or control temperature (as in the original habitat at 27 °C), a moderately high temperature (expected temperature by 2050 at 30 °C), a high temperature (expected temperature by 2100 at 32 °C) and extremely high temperatures (33, 34, 35 and 36 °C). All experiments were at a humidity of 80–90% RH. Each temperature experiment was carried out in an incubator where 30 Collembola individuals were placed in a plastic box (210 individuals in total for all seven tested temperatures). Survival was checked every 12 h for seven consecutive days and then monitored for a further seven days (14 days total). A single replication was conducted for each temperature due to the limited number of individuals, except at 36 °C, where two replicates were performed because many individuals died rapidly during the first stage. The behavior (locomotion, feeding, egg laying) of individuals was recorded, and the status of the Collembola was considered alive if they exhibited movement in response to a gentle touch with a brush. The LT_50_ or 50% mortality was tested in each experiment. Only the rearing temperatures at which all or the majority of individuals survived throughout the entire heat tolerance experiment (>80% survival rate after seven days) were selected for the next experiment: life history study.

(2) Life history study

Rearing temperatures with a survival rate higher than 80% after seven consecutive days (these were at 27, 30 and 32 °C) were selected to continually test the reproductive capacity and life history of *C. microphthalma* **sp. nov.** Food was still provided *ad libitum.* The ninety largest individuals (assumed to be adults) were taken from the stock population (all were taken from the same/similar generation). At each tested temperature, 30 individuals were used, and they were randomly paired to see if they could mate and produce eggs and offspring (F1). Each pair was placed and reared in a small square plastic box (3 × 3 × 2 cm). A total of 15 pairs were used to test at each experimental temperature (27, 30, and 32 °C). To confirm that parthenogenesis did not exist in *C. microphthalma* **sp. nov.**, it was hypothesized that if a pair of Collembola did not lay any eggs, they would be the same gender (male–male or female–female or subadult stage) and then the gender was confirmed and rechecked later under a microscope after slide preparation. At each experimental temperature, if egg(s) were present, the number of eggs laid each time and the duration of egg production at each cycle was counted. For each pair (that produced egg(s)), 3–4 eggs were randomly kept and placed in a new small square plastic box (one egg per box) to separately culture them at the same temperature as parents and for observation. A total of 30 eggs (30 boxes) (from many random pairs for each temperature) were used for studying the life history and to investigate the time duration and molting process from the egg to the adult stage at each temperature. The duration of the developmental rate of Collembola at each temperature (from egg to adult) was recorded and observed carefully by visual monitoring under a microscope. The remaining populations from each experimental temperature (27, 30, and 32 °C) were pooled in the same circular plastic box (diameter = 14 cm, height = 6 cm) and continued to cultivate for mass culture experiment at the same tested temperature as the stock population.

### 2.3. Data Analyses

The survival rate for each thermal tolerance experiment was depicted as graphs. A boxplot was used to compare the distribution of numeric data values, the mean duration of egg production and the number of eggs laid at different temperatures. The developmental rate of each experiment from egg to adult stage was compared, and significance was tested by the analysis of variance (One-Way ANOVA). All the analyses were conducted using R v.3.3.3.

## 3. Results

### 3.1. Taxonomy


**Class Collembola Lubbock, 1870**



**Order Entomobryomorpha Börner, 1913**



**Family Entomobryidae Tömösváry, 1882**



**Subfamily Entomobryinae Schäffer, 1896**



**Genus *Coecobrya* Yosii, 1956**



***Coecobrya microphthalma* sp. nov. Manee and Jantarit, 2025**


**Type material.** Holotype: female on slide. Thailand, Saraburi province, Muak Lek district, Lamphaya Klang subdistrict, Tham Dao Khao Kaeo (tham = “cave” in Thai), altitude 476 m a.s.l., 14°52′31.39″ N 101°20′16.44″ E, 3 September 2020 S. Jantarit, K. Jantapaso, A. Nilsai and K. Sarakhamhaeng leg. (sample # THA_SJ_SRI09), dark zone of the cave by the aspirator.

**Paratypes:** same data as holotype, 10 specimens (on slides, two females and eight subadults), and hundreds of specimens used in breeding experiments.

**Material deposit**: holotype and six paratypes on slides (one female, five subadults) deposited at the Princess Maha Chakri Sirindhorn Natural History Museum (NHM-PSU), Prince of Songkla University. Four paratypes on slides (one female, three subadults) deposited at the Muséum national d’Histoire naturelle (MNHN), Paris, France.

**Description.** Habitus (Figure 2A,B) Large size Entomobryidae. Body length (including head) 1.7–2.1 mm (holotype 1.9 mm). Eyes 1+1, small, in a black eye patch. Color: whitish with scattered orange pigment. Body slender, not humped nor bent at the level of Th. II.; antennae, legs and furca rather long. Th. II slightly longer than Th. III; Abd. IV about 6.5–7.5 times as long as Abd. III along the dorsal midline (N = 10).

**Pseudopores** (Figure 4B, Figure 5B–F and Figure 6A,B). Pseudopores present as round flat disks, smaller than mac sockets (Figure 4B), except for the coxae and manubrium where psp are as large as mac sockets. Pseudopores are present on many parts of the body: head, antennae, tergites, coxae and manubrium. On the head, one psp is located externally on each peri-antennal area (Figure 4B). On antennae, psp is located ventro-apically between the tip of antennal segments and the chaetae of the apical row or just below the apical row of chaetae (1–2 on Ant. I, 0–2 on Ant. II, and 2–3 on Ant. III). On tergites, 1+1 psp close to the axis from Th. II to Abd. IV (Figure 5A–F). Coxae II and III with 2–3 and 1 psp, respectively, are located close to longitudinal rows of chaetae (Figure 6A). On manubrium, 2 dorso-apical psp on each manubrial plaque (Figure 6H).

**Figure 3 insects-16-00080-f003:**
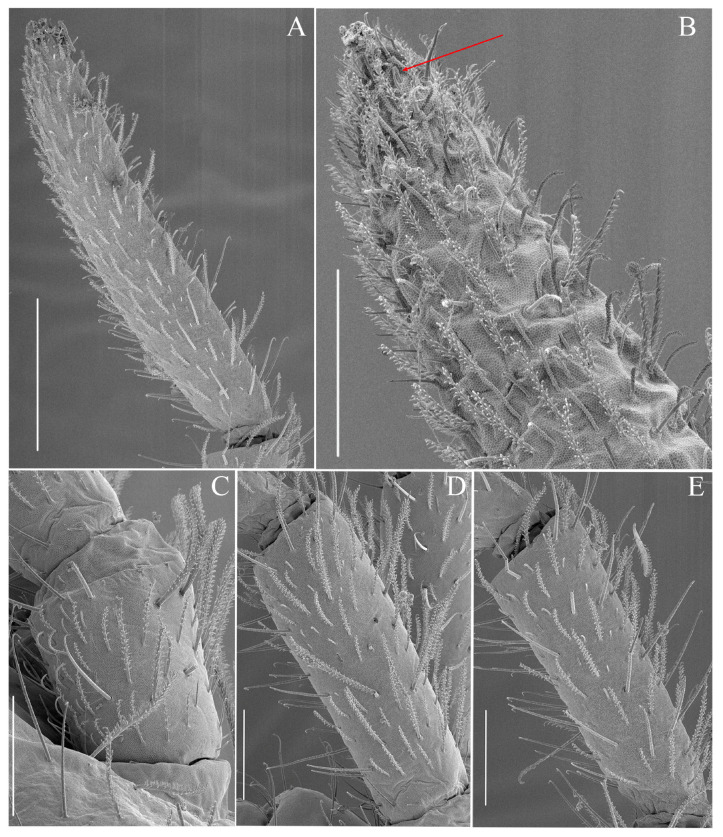
*Coecobrya microphthalma* **sp. nov.** Antennal segments and chaetae**,** (**A**) Ant. IV; (**B**) Tip of Ant. IV and subapical organite (arrow); (**C**) Ant. I laterally; (**D**) Ant. II; (**E**) Ant. III. Scale bar: (**A**) = 100 μm, (**B**) = 40 μm, (**C**–**E**) = 50 μm (SEM images).

**Figure 4 insects-16-00080-f004:**
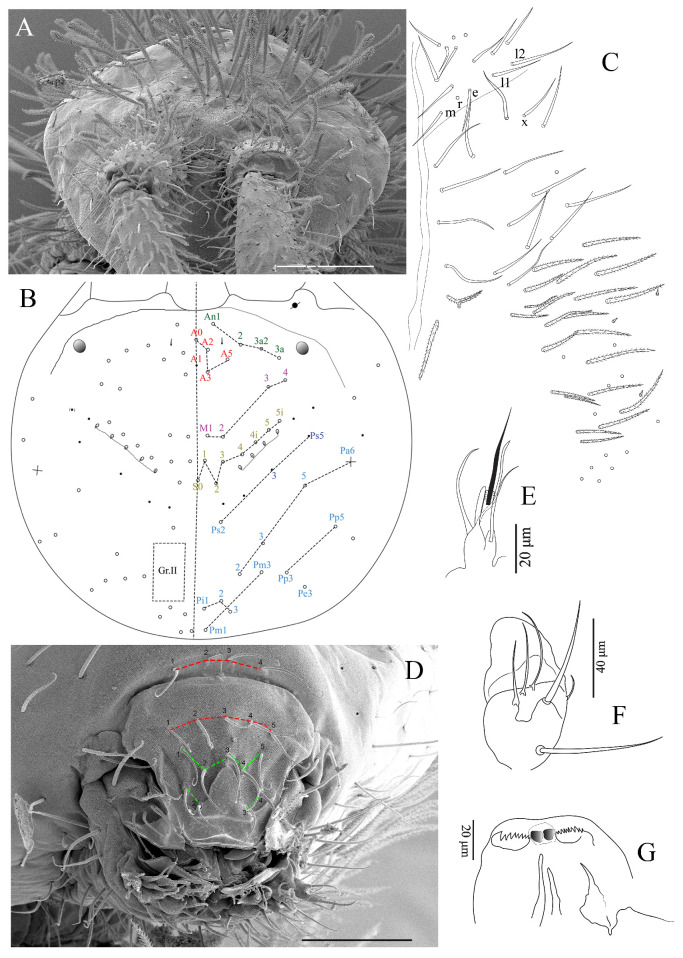
*Coecobrya microphthalma* **sp. nov.** (**A**) Dorsal head and antenna under SEM; (**B**) Dorsal head chaetotaxy; (**C**) Ventral head chaetotaxy; (**D**) Labrum and prelabral chaetae and mouthparts; (**E**) labial papilla (**E**); (**F**) maxillary outer lobe; (**G**) Ventro-distal complex of labrum. Scale bar: (**A**) = 100 μm, (**D**) = 50 μm ((**A**,**D**): SEM images).

**Figure 5 insects-16-00080-f005:**
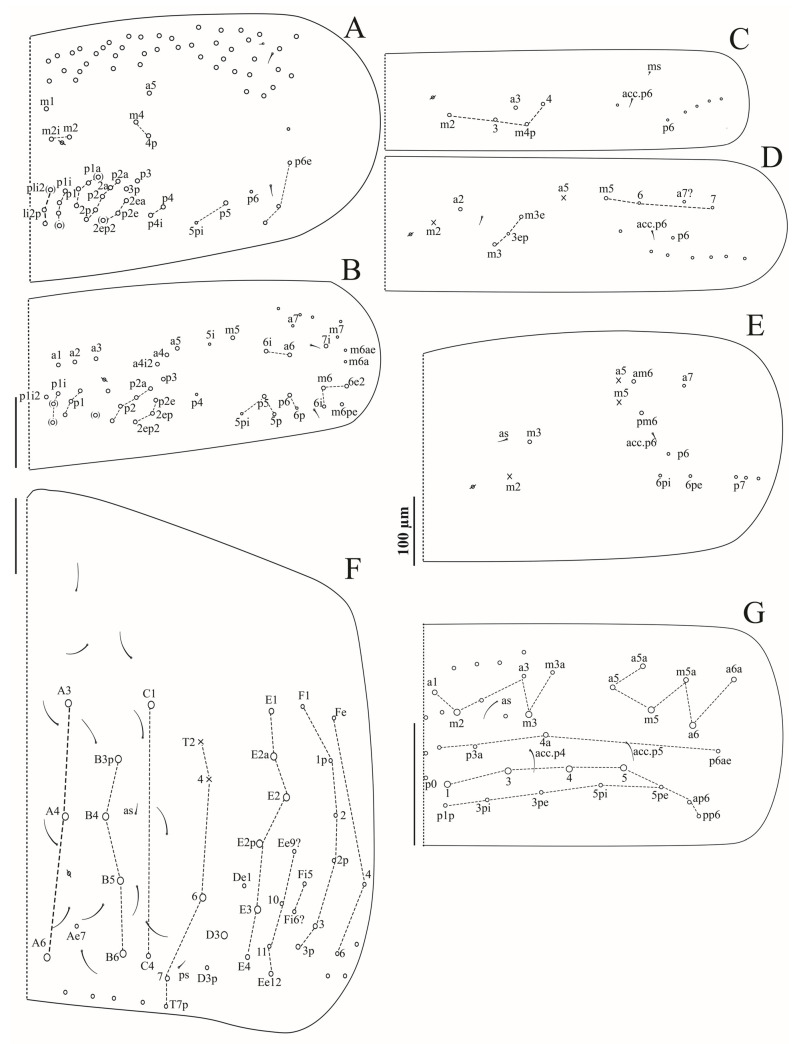
*Coecobrya microphthalma* **sp. nov.** dorsal chaetotaxy (right side): (**A**) Th. II; (**B**) Th. III; (**C**), Abd. I; (**D**), Abd. II; (**E**), Abd. III; (**F**), Abd. IV; (**G**) Abd. V; Scale bars: 100 µm.

**Figure 6 insects-16-00080-f006:**
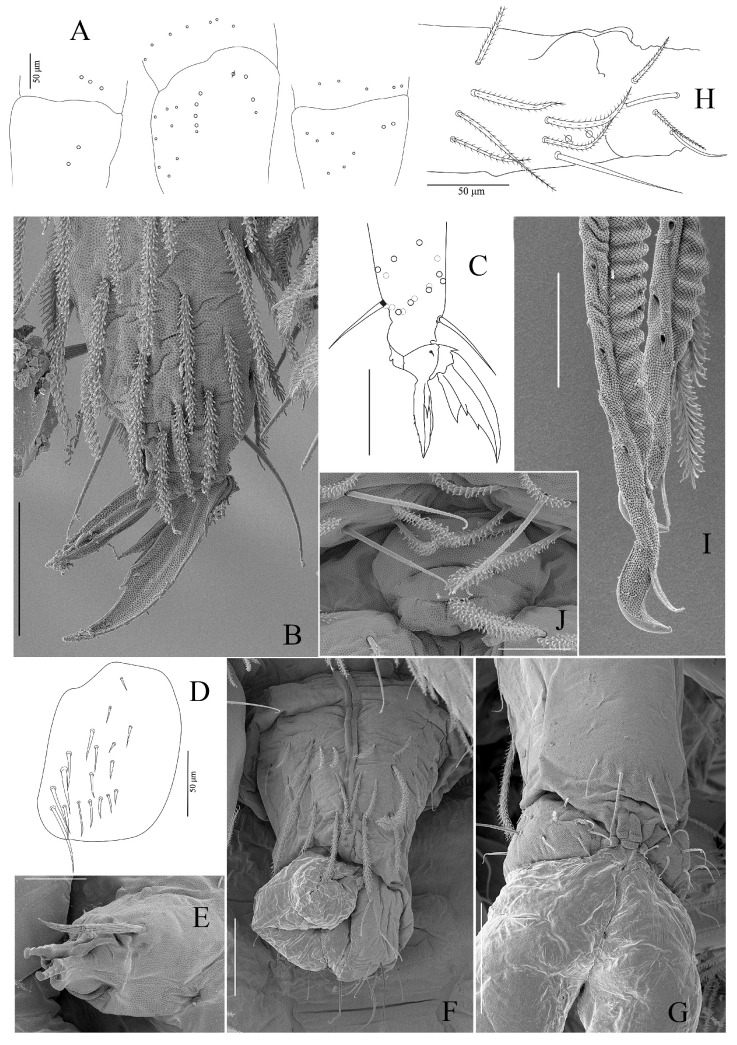
*Coecobrya microphthalma* **sp. nov.** (**A**) Coxa I–III (left to right); (**B**,**C**) Claw III morphology; (**D**) trochanteral organ; (**E**) tenaculum and a postero-basal strong serrated chaeta; (**F**), Anterior face of ventral tube; (**G**), posterior face of ventral tube and lateral flap; (**H**) Manubrium plaque; (**I**) Mucro; (**J**) Female genital plate, Scale bars: (**B**) = 30 µm; (**A,C,D,F**,**G,H**) = 50 µm; (**E**,**I**,**J**) = 20 µm ((**B**,**D**–**F**,**G**,**I**,**J**): SEM images).

Clypeus and mouthparts (Figure 4A,D–G). The clypeal region with three long, smooth prefrontal chaetae, seven middle chaetae (2+2 long smooth chaetae and three mics), and two long, smooth lateral chaetae. Distal border of the apical non-granulated area of the labrum with a relatively narrow, median V-form intrusion into the granulated area dorsally; apical edge without spines, lacking clear papillae (Figure 4D). The ventro-distal complex of the labrum is well differentiated and asymmetrical, with 1+1 distal combs of 10–12 min teeth on the right side and 9–10 strong and larger teeth on the left side, and an axial pair of long sinuous tubules (Figure 4G). Maxillary outer lobe with one basal and one apical chaetae (basal chaeta slightly thicker than the apical one) and four smooth sublobal hairs (Figure 4F). Prelabral and labral chaetae 4/5, 5, and 4 are all thin and smooth; three median chaetae are slightly longer than the two lateral ones (Figure 4A,D). Labial palp with 0, 5, 0, 4, 4 guards for papillae A–E. The lateral process of labial palp is subcylindrical, as thick as normal chaetae, with a tip beyond the apex of the labial papilla (Figure 4D,E). The mandible apex is strong and asymmetrical (right with five teeth, left with four).

Antennae (Figure 2A–C and Figure 3A–E). Antennae long, approximately 2.0–2.7 times as long as the cephalic diagonal (average = 2.48) (Figure 2A,B). Antennal segments ratio as I:II:III:IV = 1:1.77:1.70:2.57 (average N = 10). Antennal segments are not annulated nor subdivided. Antennal chaetal types are not analyzed in detail. Smooth spiny mic at the base of antennae: 3–4 dorsal, three ventral on Ant. I; one external, one internal and one ventral on Ant. II; smooth, straight, long chaetae on antennae present in all segments. Ant. I ventrally with many subcylindrical, hyaline sens in its middle to apical part and many long, smooth, straight chaetae (Figure 3C). Ant. II without paddle-like chaetae, many smooth, long and straight chaetae on both dorsal and ventral sides and many subcylindrical, hyaline sens (Figure 3D). Ant. III with many smooth, long, thick and straight chaetae both on dorsal and ventral sides, and many subcylindrical, hyaline sens (Figure 3E). Ant. III organs with typical five sens; sens 1 and 4 subequal, hyaline; sens 2 and 3 swollen; sens 5 acuminate, dark and shorter. Ant. IV long, without apical bulb, not subdivided. Smooth, long, thick and straight chaetae are present at the base of Ant. IV of both sides (Figure 3A,B). The subapical organite is weakly swollen, slightly enlarged apically, and inserted dorsally (Figure 3B).

Dorsal head chaetotaxy (Figure 4A,B). Dorsal cephalic chaetotaxy with four antennal chaetae (An1–2, An3a2, An3a; An3 as mes), four anterior chaetae (A0, A2–3 and A5), A1 as mic, four median mac (M1–4) and eight sutural mac (S0, S1–4, 4i, 5, 5i), Gr. II without mac; 4–5+4–5 scale-like structures below the sutural mac present (sensu Jantarit et al., 2019 [3]), inside the integument (Figure 4B), (scale-like structures not detected under SEM); a pair of short cephalic trichobothria, external and close to the middle of the head (Figure 4B).

Ventral head chaetotaxy (Figure 4C). Chaetae of labial basis all smooth (mrel1l2) chaetae m, e and l1 subequal, r as mic, l2 longest. Postlabial chaetae X, X3 and X4 as smooth chaetae; X2 absent. The cephalic groove with six long chaetae on each side, the four anterior smooth the two posterior ciliated.

Tergites (Figure 5A–G).

Th. II with three (m1, m2, m2i) medio-medial, three (a5, m4, m4p) medio-sublateral and 21–24 posterior mac; 1+1 ms and 2+2 sens laterally (Figure 5A).

Th. III with 29–32 mac, a6, p5–6 as mac, 2+2 sens laterally (Figure 5B).

Abd. I with five (a3, a4, m2–4, m4p) central mac, 1+1 ms and 1+1 sens laterally (Figure 5C). Abd. II with three (a2, m3, m3e) central and one (m5) lateral mac, 2+2 tric without modified surrounding chaetae, 1+1 sens laterally and 1+1 sens above m3 (Figure 5D).

Abd. III with one (m3) central and two (am6, pm6) lateral mac. 3+3 tric without modified surrounding chaetae, 1+1 sens laterally, 1+1 sens near m3 (Figure 5E).

Abd. IV with eight central mac (A3, A4, A6, B3p, B4–6, C1), and seven lateral mac (T6, D3, E1, E2a, E2, E2p, E3), 2+2 tric and about 13 long S-like chaetae, without modified chaetae (Figure 5F).

Abd. V with at least eight obvious mac (a6, m2–3, m5, p1, p3–5), several mes to small mac, and 3+3 sens (Figure 5G).

Abd. VI not analyzed.

S-chaetae formula from Th. II to Abd. V: 2+ms, 2/1+ms, 2, 2, ≈13, 3; ps and as sens on Abd. IV 1/3 as long as S-like chaetae (Figure 5A–G).

Legs (Figure 2A–B, Figure 6A–D). Legs long; the tita of leg III is slightly longer than the tita of legs I and II (Figure 2A,B). Legs devoid of scales, covered with ordinary chaetae of various lengths and mostly ciliated, and an inner row of 3–4 smooth chaetae on tita. Trochanteral organ with 18–24 smooth, straight, unequal spine-like chaetae (Figure 6D). Tibiotarsi II and III with one large inner ciliate mac at 1/3 from the basal side. The distal whorl of tita with 10 subequal ciliated mes, irregularly arranged, and a dorso-apical tenent hair, usually pointed (Figure 6B,C). Absence of smooth, thin and long chaeta close to the tenent hair (as observed in *C. sirindhornae*). Ventro-distal smooth chaeta of tita III thick, pointed, erected, rather long. Claw slightly elongated. Unguis with a tiny unpaired inner tooth, a pair of subequal basal teeth at about 52–56% of the inner edge from basis, and a tooth on the outer edge at about 15% from basis. Unguiculus long, with minute outer teeth, approximately ⅔ long as the inner edge of the claw, rather narrow, not swollen baso-internally, pointed apically (Figure 6B,C).

Ventral tube (Figure 6F,G). The ventral tube is about three times longer than wide. Anterior face with 7–10+7–10 ciliated chaetae, sometimes asymmetrically arranged, 2–3 of them larger than the others (Figure 6F); posterior face with 1+1 long smooth and 1+1 smooth distal chaeta, and 2–3 mic (Figure 6G). Lateral flaps with 8+8 rather long, smooth and thin chaetae, with sometimes a few ones larger than others (Figure 6G).

Furcal complex (Figure 6E,H–I). Tenaculum with four large teeth of decreasing sizes from the basal to the distal one on each ramus, on a prominent, irregular corpus, with a postero-basal strong serrated chaeta bent distally (Figure 6E). Mucrodens 1.95–2.50 times longer than the manubrium (N = 10). Manubrium densely covered by ciliated chaetae both dorsally and ventrally with rows of 9–11+9–11 smooth chaetae dorsally. Ventro-distal part of the manubrium with 10+10 to 12+12 ciliated chaetae. Manubrial plaque with two pseudopores each, accompanied by two ciliated chaetae and one smooth chaeta (Figure 6H). Dens annulated, without spines and covered with ciliated chaetae, except on its dorsal side. Dens with 2+2 smooth chaetae latero-basally. Distal smooth region of dens slightly longer than the mucro. Mucro falcate and strong, basal spine long, nearly reaching the tip of the mucronal tooth (Figure 6I).

Genital plate. Female genital plate with 2+2 genital mic (Figure 6J), male genital plate not clearly observed.

**Ecology.** *Coecobrya microphthalma* **sp. nov.** was collected in the dark zone of Tham Dao Khao Kaeo on the cave floor about 100–120 m from the main entrance. This species was found mainly in mesotrophic habitats where guano and organic matter were often present. The air temperature in the cave fluctuates locally from 25.4 to 28.0 °C, while the chamber where the specimens were collected was at 26.4–27.2 °C. The soil temperature was 23.5–24.8 °C with a humidity of 81–87%. The cave is located in a rather massive limestone hill and there are 160 steps leading up to the gated entrance. Tham Dao Khao Kaeo is 442 m long and is developed as a temple cave tourist attraction. Although artificial lights, shrines and Buddha images have been installed in the cave, this new species was mostly observed and collected in a large dark chamber.

**Etymology.** *Coecobrya microphthalma* **sp. nov.** is derived from the Latin meaning “having small eyes”.

**Remarks.***Coecobrya microphthalma* **sp. nov.** belongs to the *boneti*-group characterized by the presence of eyes. The new species has 1+1 small eyes like six other species of the same group (C*. boneti* (Denis, 1948) [28], *C. sanmingensis* Xu and Zhang, 2015 [40], *C. indonesiensis* (Chen and Deharveng, 1997) [41], *C. tukmeas* Zhang, Deharveng and Chen, 2009 [6], *C. oculata* Zhang, Bedos and Deharveng, 2016 [34] and *C. chompon* [4]). Among Thai cave species, *Coecobrya microphthalma* **sp. nov.** is most similar to *C. chompon* in having relatively long antennae, labial chaetae as mrel1l2, presence of long smooth straight chaetae on antennae, 3 medio-medial mac on Th. II, 3 central mac on Abd. II, 1 central mac and ms present on Abd. III, 3 ungual inner teeth. However, the new species is morphologically different from *C. chompon* by its larger body size (the largest size for the *boneti*-group so far), the presence of orange pigment on the body, sublobal hair on the maxillary outer lobe with 4 (vs. 3) chaetae, Th. II with 2 (vs. 3) medio-sublateral mac, Abd. I with 5 (vs. 6) mac, Abd. III with 2 (vs. 3) lateral mac; Abd. IV with 8 (vs. 7) central mac; the anterior face of the ventral tube with 7–10 (vs. 5–6) ciliated chaeta, and mucronal spine almost reaching mucronal apex (vs. not reaching mucronal apex). The morphological comparison of all known species of the *boneti*-group is given in Table 1.


**Key to world species of *Coecobrya* of the *boneti*-group**


1. Eyes > 1+1……………………………………………………………………………….…2

- Eyes 1+1…………………………………………………………………………………….4

2. Eyes 2+2; Th. II with 2 medio-sublateral mac; Th. III with 20 mac; Abd. III with 2 lateral mac and pointed tenent hair………………………………..*C. tetrophthalma* **(Vietnam)**

- Eyes 3+3; Th. II 3 medio-sublateral mac; Th. III with 22–23 mac; Abd. III with 3 lateral mac and clavate tenent hair……………………………………...………………………3

3. Violet-bluish color; labial basis as “mrel1l2”; Th. II with 3 medio-medial mac; Abd. I with 5 mac; Abd. II with 3 central mac; Abd. III with 1 central mac; Abd. IV with 3 central mac; ms on Abd. III present ………………………………………… *C. mulun* **(China)**

- White color; labial basis as “mRel1l2”; Th. II with 1 medio-medial mac; Abd. I with 6 mac; Abd. II with 4 central mac; Abd. III with 2 central mac; Abd. IV with 7 central mac; ms on Abd. III absent ………………………………………….................*C. qin* **(China)**

4. Dorsal side of head with 4 mac in the “M” series and Gr. II with 4–5 mac; sublobal hairs on the maxillary outer lobe with 4 chaetae; Abd. IV with 8 central mac… *C. microphthalma* **sp. nov. (Thailand)**

- Dorsal side of head with 3 mac in the “M” series and Gr. II without mac; sublobal hairs on the maxillary outer lobe with 3 chaetae; Abd. IV ≤ 7 central mac………………………5

5. Dorsal side of the head with 5 chaetae in the “An” series; Th. II with 4 medio-medial mac; Abd. IV with 4 central mac …………………………………………….*C. oculata* **(China)**

- Dorsal side of the head with 4 chaetae in the “An” series; Th. II with 3 medio-medial mac; Abd. IV with 6–7 central mac ………………………………………………………..……6

6. Abd. I with 4–5 mac; Abd. II with 3 mac ……………………………………………….7

- Abd. I with 6 mac; Abd. II with 4 mac …………………………………………………...8

7. Labial basis as “mRel1l2”; chaetae along the cephalic groove with 7–8 chaetae; Th. II with 3 medio-sublateral mac; posterior mac of Th. II with 15; Abd. I with 5 mac; Abd. III with 3 central mac and 2 lateral mac; Abd. IV with 7 central mac………… *C. sanmingensis* **(China)**

- Labial basis as “mrel1l2”; chaetae along the cephalic groove with 3 chaetae; Th. II with 2 medio-sublateral mac; posterior mac of Th. II with 18–20; Abd. I with 4 mac; Abd. III with 1 central mac and 3 lateral mac; Abd. IV with 6 central mac………… *C. tukmeas* **(Cambodia)**

8. Abd. IV with 6 central mac; Gr. II of dorsal side of head with 4 chaetae … *C. boneti* **(Vietnam)**

- Abd. IV with 7 central mac; Gr. II of dorsal side of head different …………….……9

9. Gr. II of the dorsal side of the head with 5 chaetae; pointed tenent hair; Th. III with 30 mac; the anterior face of the ventral tube with 10 ciliated chaetae………...…*C. indonesiensis* **(Indonesia)**

*-* Gr. II of dorsal side of head with 3 chaetae; clavate tenent hair; Th. III with 26 mac; the anterior face of the ventral tube with 5–6 ciliated chaetae………….……*C. chompon* **(Thailand)**

### 3.2. Effect of Temperature on the Cave Collembola Coecobrya microphthalma **sp. nov.**

#### 3.2.1. Survival Rate

The results revealed that the survival rate clearly decreased when the temperature increased (Figure 7). *Coecobrya microphthalma* **sp. nov.** cannot maintain its population at 34, 35 and 36 °C after seven days of the experiment (below the LT_50_). At the extremely high temperatures of 35 °C and 36 °C, populations of *C. microphthalma* **sp. nov.** sharply dropped and reached 100% mortality in the laboratory test. At 34 °C, the number in the population dramatically declined to less than 50% (LT_50_) on day 7. When the culture was expanded to 14 days it was clearly shown that the number in the population continued to drop and could not survive to the end of the experiment. At 33 °C, *C. microphthalma* **sp. nov.** could sustain its population at more than 50% (> LT_50_) for the first seven days. However, when the culture experiment continued for 14 days, the proportion of surviving specimens in the population continued to decrease. The optimal temperature for sustaining the population of *C. microphthalma* **sp. nov.** was 27 °C (the control temperature, which is equal to equivalent to the cave temperature). At 30 °C, the survival rate dropped to 57%, and below at higher temperatures. At 32 °C, less than 50% of specimens remained alive (LT_50_) after 10 days of culture, but the survivors were still active and could produce eggs in our experiment. Consequently, cultured experiments at 27, 30, and 32 °C were selected to study the life history of *C. microphthalma* **sp. nov.** in the next experiment described below.

#### 3.2.2. Life History of *Coecobrya microphthalma* **sp. nov.**

(1) Reproductive capacity

In our study on reproductive capacity, a total of 15 random pairs of *C. microphthalma* **sp. nov.** were used for each experiment. There were five pairs at 27 °C, eight pairs at 30 °C and eight pairs at 32 °C that successfully produced eggs. The reproductive capacity of *C. microphthalma* **sp. nov.**, which included egg production and the number of egg-laying days, differed significantly depending on the temperature. The duration of egg production and the number of egg-laying days significantly decreased when the temperature increased (*p* < 0.001; see Figure 8). The average duration for egg-laying was 6.55 ± 0.18 days (n = 20) at 27 °C, 4.80 ± 0.44 days (n = 10) at 30 °C and 3.50 ± 0.34 (n = 10) days at 32 °C (Figure 8). Meanwhile, the number of eggs laid at each temperature differed significantly, with an average of 28.00 ± 2.50 eggs at 27 °C (n = 20), 21.60 ± 2.14 eggs at 30 °C (n = 10), and 16.50 ± 1.33 eggs at 32 °C (n = 10) (*p* < 0.01) (Figure 8). Note: N = number of observations on egg-laying days as females can lay eggs several times during their life, and some pairs also died during the experiment.

(2) Development rate

The comparison of egg hatching time and the duration of postembryonic development of *C. microphthalma* **sp. nov.** (molting of the juvenile until emergence of the adult) were determined and observed at temperatures of 27, 30, and 32 °C. The results revealed that *C. microphthalma* **sp. nov.** continued to grow from the 1st instar juvenile and required six molts to reach the adult stage (N = 30). The gender was confirmed under a microscope after slide preparation. There is a statistical difference for each temperature experiment between the egg hatching time to the 1st instar juvenile and from the 1st instar juvenile to the 6th instar juvenile (*p* < 0.001, Table 2, Figure 9). The average time interval from egg hatching to reaching the 1st instar juvenile was 172.80 ± 1.78 h at 27 °C, 71.03 ± 0.43 h at 30 °C, and 60.93 ± 3.37 h at 32 °C (n = 30). The duration of the molting process from 1st instar to 6th instar varied and differed significantly among the tested temperatures (*p* < 0.001). The average time to reach adult stage was 73.64 ± 2.57 h at 27 °C, 51.91 ± 1.83 h at 30 °C, and only 47.34 ± 1.01 h at 32 °C (Table 2). The average time interval of each development stage is summarized in Table 2, with significant statistical differences. The results show that the higher the temperature, the faster the time to reach the adult stage. An average time of only 233.53 h (9.73 days) at 32 °C was spent to reach the adult stage but required 257.04 h (10.71 days) at 30 °C and 370.3 h (15.44 days) at 27 °C. Overall, at the original environment temperature (27 °C), *C. microphthalma* **sp. nov.** needed at least 543.43 h (22.64 days) to reach the adult stage from the time of egg production. If the temperature increased by 3 °C (to 30 °C), it required an average time of 328.07 h (or 13.67 days) for the whole process, and there was an even greater decrease in the maturation process when the temperature increased by 5 °C (to 32 °C), taking only 294.46 h (12.27 days) to complete its life cycle.

## 4. Discussion

Insects are poikilotherm organisms whose body temperature is regulated by the surrounding environment. Temperature has long been known as the principal physical factor that directly affects metabolism and activity, life history, population dynamics and distribution of insects [44,45,46,47]. Normally, insects prefer to live in an optimum temperature range. If the ambient temperature drops, the rate of insect development and growth will slow down and pause at the lowest critical temperature. On the other hand, if the temperature rises, the rate of insect development and growth will increase until it reaches the highest critical temperature. If the ambient temperature continues to rise, the rate of insect development and growth is reduced and ceases (dies) at temperatures above this threshold [48,49,50]. In this study, it is shown that a slight increase in temperature in the first experiment (27 to 30 °C) appears to have a limited effect on the life history of *C. microphthalma* **sp. nov.** An increase in temperature of 5 °C from the control condition (27 °C to 32 °C) obviously accelerates the process of postembryonic development with statistical significance (*p* < 0.01). This is expected as higher temperatures increase the metabolic and growth rate but decrease the life span of both subterranean and edaphic collembolan species [16,51,52]. In our results, temperature also plays a role in egg production by *C. microphthalma* **sp. nov.** by illustrating that when the temperature increases by 3 °C and 5 °C from the control temperature (27 °C), the number of eggs produced is reduced significantly (*p* < 0.01). This scenario is coincident with the previous studies about poduromorph springtails living in subterranean habitats [16], as well as about hemiedaphic Onychiuridae Collembola [53]. This suggests that the energy use of Collembola is partially relocated to other processes than reproductive investment when temperature increases.

Considering the LT_50_, if the temperature is greater than 32 °C (5 °C above the control temperature), *C. microphthalma* **sp. nov.** tends to cease its activities, and the rate of mortality strongly increases, indicating that the temperature has exceeded a threshold. The population of *C. microphthalma* **sp. nov.** gradually declines in these conditions and could not survive after 1 or 2 weeks in the laboratory tests (Figure 7) due to physiological limitations. It is well known that higher temperatures directly cause insects to dehydrate, face protein denaturation and cellular damage, and affect enzyme structure, including enzyme malfunction [54,55], resulting in the insects’ cessation of activity and eventual death.

At 32 °C (5 °C above the control), *C. microphthalma* **sp. nov.** was able to withstand the heat experiment in the early stages and then the population began to decrease dramatically in the second week. However, many individuals (N = 13) of the F0 generation survived and were active in the 32 °C experiment; they were even able to lay eggs that could successfully hatch to the adult stage. The F1 generation was also fully active. However, surprisingly, all F1 populations (N > 30) could not continue to produce eggs and offspring in our mass culture experiment. Long-term exposure to heat apparently affects the fertility of *C. microphthalma* **sp. nov.**. This is also illustrated in the case of the flour beetle *Tribolium confusum*, where exposure to high temperatures, especially during embryonic development, caused females to become sterile to varying levels depending on the degree and duration of exposure to high temperatures [56]. Also, in this species, exposure to elevated temperatures reduces sperm quantity and impairs the reproductive function of male beetles [57], including males of the fungus gnat [58]. Exposure to high temperatures even induces some male insects, such as flies, to change their gender [59,60] as well as to promote sex ratio imbalance upon emergence [61,62]. It was not the case in our experiments, as we checked the sex of the specimens after egg laying, and found males as expected. These convergent results in different arthropod groups are still very limited but call for deeper investigations as potential opportunities to detect the impacts of global warming. In this respect, tropical cave species that are not dependent on temperature fluctuations may provide good biological models.

The most concerning issue is that 32 °C, which is 5 °C higher than the original cave habitat of the studied species, is at the upper limit of the range of temperature predicted by the IPCC (2023) [17] for 2081–2100. If this rise of 5 °C in temperature comes true, it would largely impact the population dynamic of tropical cave species like *C. microphthalma* **sp. nov.**, which would be put at a high risk of extinction by the end of the century unless it can migrate to less warm habitats or adapt to the rising temperatures over time. Caves maintain a nearly constant temperature throughout the year, which is, at low altitudes and in the absence of particular sources of heat, almost the average annual temperature, according to a huge amount of data [63]. The temperature in caves will, therefore, reflect a large extent of outside temperature in the global warming dynamics, evidence corroborated by theoretical models [64,65]. Actually, global climate alterations are currently modifying subterranean microclimates worldwide as they did in the past, even if hard data are lacking for the tropics. Changes in underground temperatures are, however, more or less delayed compared to those experienced in surface environments, depending on air circulation, hydrology, presence of ice, and presence of bats, but available archives usually point to rather short lags, though it may require up to a few decades in special cases [65]. In any case, subterranean environments will be subjected to the effects of climate change [66,67].

The genus *Coecobrya* includes some of the most morphologically modified species in caves, aside from unmodified species. The genus can be categorized into three groups according to its degree of morphological adaptation: non-troglomorphic, troglomorphic and highly troglomorphic [3]. The new species described here can be qualified as moderately troglomorphic, having reduced eyes (1 + 1) and pigmentation and slightly elongated antennae and claws. The troglomorphic Collembola are troglobionts, i.e., stenothermal animals that live or survive within a narrow and constant temperature range and in a constantly saturated atmosphere. They generally have smaller distribution ranges than surface species and are sensitive to temperature fluctuations, but to an unknown extent, as environmental conditions are not the primary driver of sensitivity to heat tolerance, according to Pallarés et al. (2019) [68]. The new cave species studied here, *C. microphthalma* **sp. nov.**, has a flexible thermal tolerance, with an optimal thermal range of +3 to 4 °C, and can complete its life history even at 5 °C above the temperature of its original habitat. Raising the temperature by 3 °C (or more) also promotes faster development stages and increases the population dynamics in this case. Our result is comparatively similar to the pioneer comparative study of Thibaud (1970) [16], who made experimentations on several cave Poduromorpha (troglobionts and troglophiles species) from temperate regions. He found that the optimal thermal range of the troglobitic species *Typhlogastrura balazuci* is narrow with an optimum of +1.5 °C from the absolute optimum, while the troglophile species of the genera *Acherontiella*, *Ceratophysella*, *Hypogastrura*, *Mesachorutes* and *Schaefferia* have broader thermal ranges from +3 to +7.5 °C from the absolute optimum temperature.

It cannot be ruled out that an exposure to nonlethal temperatures, though it seems less worrying for *C. microphthalma* **sp. nov.**, may incur “hidden costs”” through sublethal effects (e.g., oxidative stress) which could influence fitness over the long term [69]. Still, these physiological impacts are difficult to measure in tiny arthropods and have not been estimated here nor by Thibaud (1970) [16]. In any case, the results of our study provide new insights into the physiological adaptation of troglobiotic Collembola in the tropics, more specially a broader thermal tolerance than expected. Subterranean habitats are characterized by very low daily and seasonal variations in temperature. The Climatic Variability Hypothesis (CVH) [22] assumes that their species are, therefore, not under selective pressure to develop physiological adaptations to extreme thermal fluctuations for survival, contrary to surface species. Raschmanová et al. (2018) [63], testing this hypothesis for cave and soil Collembola, found that these species have specialized adaptations to withstand fluctuating environmental conditions, supporting the climatic variability hypothesis. The rule established from temperate cave studies [16,63] is that cave-obligate species often exhibit traits such as slower metabolism, extended life cycles, and a more gradual development process to cope with the harsh and resource-scarce environments of caves. In contrast, non-cave species, which typically inhabit more dynamic environments with access to greater resources, tend to develop more quickly to reach maturity [70]. The result of the present study on *C. microphthalma* **sp. nov.** may, however, depart from this rule. The very short length of its postembryonic development comes as a surprise. The complete life cycle of *C. microphthalma* **sp. nov.**, which lasts about 22 days in its original temperature climate, is exceptionally short (equating to 16 generations per year) and unexpected, but comparison with surface Collembola under the same tropical climate is lacking. This highlights the necessity of comparing the life cycles of tropical surface-dwelling Collembola to evaluate whether the CVH hypothesis is applicable to tropical species.

The life cycle of cave-dwelling collembola can vary depending on the groups or species, degree of adaptation, habitat structure and geographical regions. Available data for a complete life cycle of cave collembola is extremely limited, but the Hypogastruridae studied by Thibaud (1970) [16], troglobionts or troglophiles, had cycles much longer than that of *C*. *microphtalma* **sp. nov.** In temperate Hypogastruridae, Thibaud (1970) [16] has shown, for instance, that developmental growth up to adult of the troglobiotic species *Typhlogastrura balazuci* Delamare-Deboutteville, 1951 required 84 days at 10.5 °C, while the troglophilic *Ceratophysella bengtssoni* (Agren, 1904) needed 51 days to reach the adult stage at 10.5 °C, i.e., near their optimal temperature. The life cycle of *C microphtalma* **sp. nov.**, an Entomobryidae, is only 22 days at optimal temperature. It is possible that this very short cycle is at least partly taxon-related, given the very large phyletic distance with Hypogastruridae. Once again, we, however, lack data for other Entomobryidae.

The scarcity of food resources in caves is known as one of the hallmarks of cave environments and a primary factor driving cave organisms to reduce their metabolism, which in turn slows their development and extends their life cycles [71]. This phenomenon appears to be particularly effective in temperate caves, especially those with an oligotrophic environment. However, in tropical regions, food resources in caves appear to be often abundant, particularly guano from bats, organic debris from the surface or seasonal flooding. This new, slightly troglomorphic species, *C. microphthalma* **sp. nov.**, is found exclusively in a big chamber within the deep and dark area of a cave in mesotrophic habitats where bat guano is frequently present. The abundant nutrients in this cave may have reduced the need for this new species to adapt to harsher cave conditions, as the environmental factors and resources are more predictable, resulting in faster-than-expected developmental growth. Thibaud (1970) [16] gives some information on the first instar length of troglophilic and guanobotic species, which are slightly but significantly shorter than those of the troglobiotic Hypogastruridae. *C. microphthalma* **sp. nov.** is possibly what has been qualified of *troglobites-guanobites,* by Decu (1986) [72], and may have also a shorten life cycle than that of full troglobionts. Again, the complete life cycle of other cave Collembola in tropical regions is unknown. Expanding the laboratory culture of tropical Collembola, ranging from the edaphic to strictly troglobiotic and troglomorphic species across different groups, will be necessary to understand the drivers of their life cycle. Several hypotheses exist, however, regarding the relationship between developmental growth, life cycle, and the characteristics of subterranean environments, and they will have to be tested when more data are available [70].

In the context of global warming, a comprehensive understanding of physiological mechanisms is crucial for predicting the biotic responses of organisms to climate change and assessing extinction risk [73]. Experimental data on thermal tolerance in Collembola are still scarce and have only been carried out in a few geographical areas, mostly from the polar and temperate regions, with very few attempts from subtropical and tropical regions [15,16,53,74]. This study is the first attempt to address this issue in tropical countries using Collembola as a model. Furthermore, the studied species is representative of the largest group of springtails that colonized caves worldwide (Entomobryomorpha). The novel findings of this study are the first contribution to exploring cave invertebrate response to potential climate changes in the tropics. Data are limited to a single species, but developing sound policies for cave protection and conservation would need much larger datasets in the same field. Species could be selected among the wealth of species that live in Thai caves [9], carrying out studies on Collembola of different groups and different ecology, in order to get an increasingly meaningful overview of the impacts to be expected, would global warming continues to increase.

## Figures and Tables

**Figure 1 insects-16-00080-f001:**
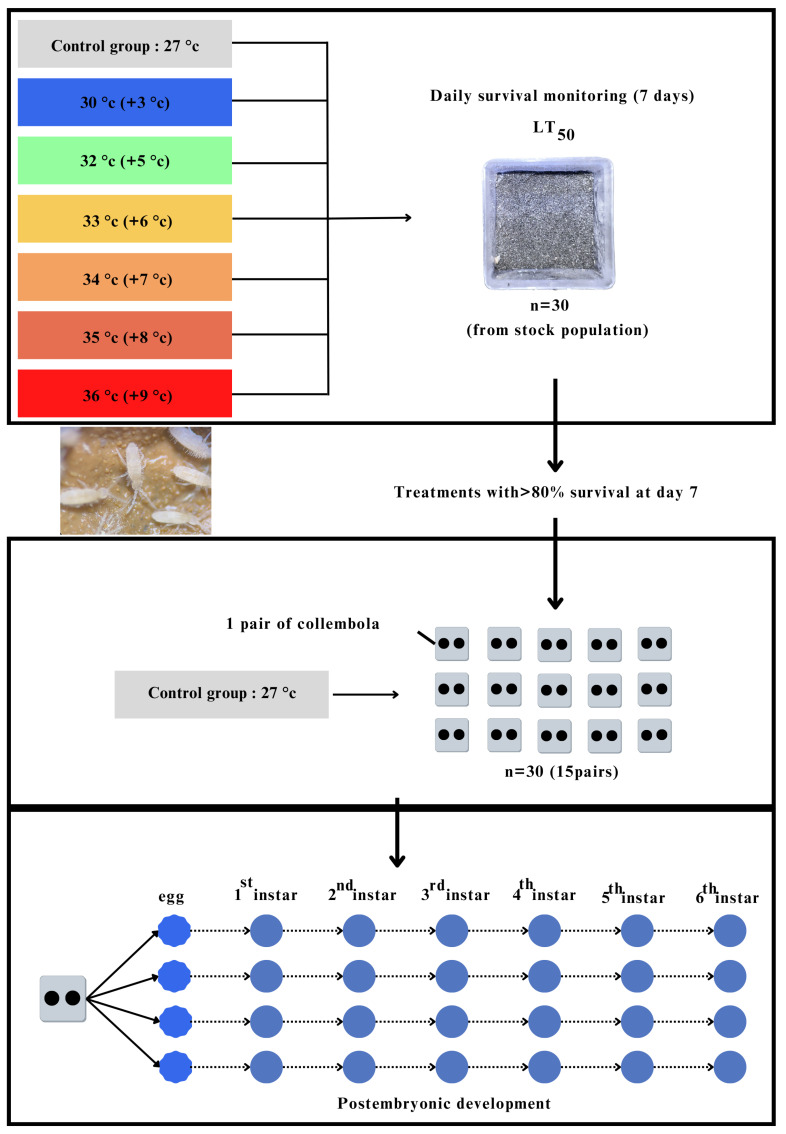
Diagram of thermal tolerance experiment for *C. microphthalma* **sp. nov.**

**Figure 2 insects-16-00080-f002:**
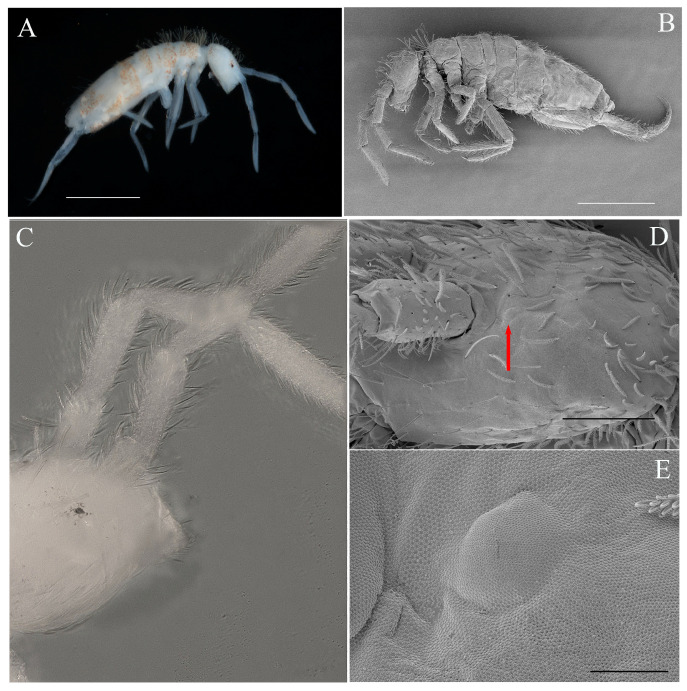
*Coecobrya microphthalma* **sp. nov.** (**A**) habitus under microscope; (**B**) habitus under SEM; (**C**) head and black eyepatch under slide; (**D**) head and eye (arrow) under SEM; (**E**) enlargement of eye under SEM. Scale bar: (**A**,**B**)= 500 μm, (**D**) = 100 μm, (**E**) = 10 μm ((**A**,**C**): microscope images; (**B**,**D**,**E**): SEM images).

**Figure 7 insects-16-00080-f007:**
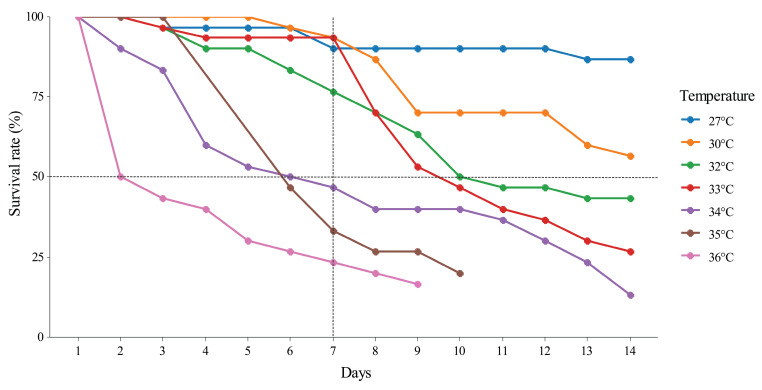
Survival rate of *C. microphthalma* **sp. nov.** at different temperatures (27 °C as control, 30, 32, 33, 34, 35, and 36 °C).

**Figure 8 insects-16-00080-f008:**
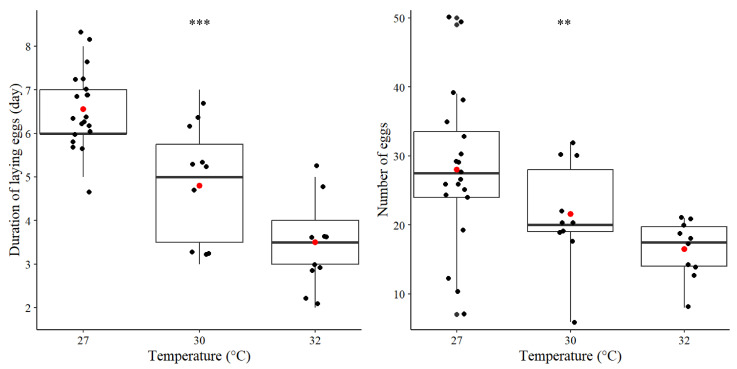
Average duration of egg production at 27, 30, and 32 °C (**left**); red dot represents an average number, and black dots of each box represent a number of observations on egg-laying days. Average number of laid eggs at 27, 30, and 32 °C (**right**); red dot represents an average number, black dots of each box represent a number of observations on a number of eggs; ** = statistic difference at *p* < 0.01, *** = statistic difference at *p* < 0.001.

**Figure 9 insects-16-00080-f009:**
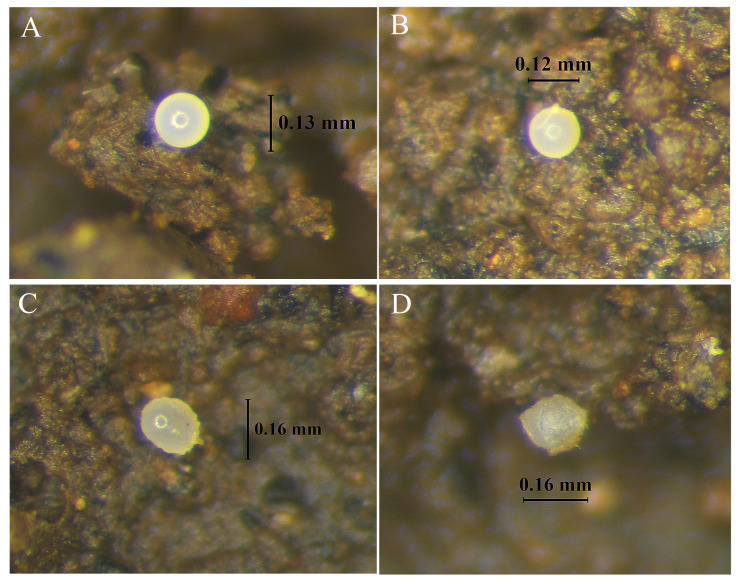
Egg size and form of *C. microphthalma* **sp. nov.** during the development process from younger to older stage (**A**–**D**) observed under light stereomicroscope.

**Table 1 insects-16-00080-t001:** Comparison of world species of *Coecobrya* from the *boneti*-group.

Characters/Species	*Coecobrya boneti*	*C. chompon*	*C. indonesiensis*	*C. microphthalma* sp. nov.	*C. mulun*	*C. oculata*	*C. qin*	*C. sanmingensis*	*C. tetrophthalma*	*C. tukmeas*
Body length (mm)	up to 1.24	1.0–1.4	≈1.44	up to 2.1	up to 1.40	up to 1.38	up to 2.09	up to 1.09	up to 1.27	up to 1.5
Color	whitish with scattered orange pigment	white	white with pale yellow pigment	white with scattered orange pigment	violet-bluish	white	white	white	whitish with scattered orange pigment and orange bands	ground color pale light orange to whitish
Eyes	1+1	1+1	1+1	1+1	3+3	1+1	3+3	1+1	2+2	1+1
Ant./head ratio	1.9–2.1	2.1–2.84	1.6–1.9	2.0–2.76	1.6–1.9	1.6–1.7	?	1.96–2.02	1.65–1.97	1.75
Long smooth straight chaetae on antennae	present	present	present	present	?	absent	absent	absent	absent	?
No. of rod-like chaetae on Ant. II	2	2–3	2	1	?	1	1	?	1	1
Clypeal chaetae										
Prefrontal	?	3s	?	3s	?	?	?	3s	?	?
Facial	?	1–2c4s	?	6s3m	?	?	12c	3c2s	?	?
Dorsal head chaetae										
An	4	4	4	4	4	5	4	4	4	4
A0	mac	mac	mac	mac	mac	mac	mac		mac	mac
M	3	3	3	4	3	3	3	3	3	3
Gr. II	4	3	5	0	4	5(4)	4	4	4	4
Sublobal hairs on maxillary outer lobe	3	3	3	4	3	?	3	3	3	3
Labial chaetae	mrel1l2	mrel1l2	mrel1l2	mrel1l2	mrel1l2	mRel1l2	mRel1l2	mRel1l2	mrel1l2	mrel1l2
Postlabial chaetae X	smooth, long	long	smooth, long	smooth, long	long	ciliate	ciliate	ciliate	smooth, long	long
Chaetae along the cephalic groove	3s	4s1–4c	5s?	4s2c	3s	2s4c	2s5c	4s4(3)c	3s	3s
Chaetotaxy of Th. II										
Medio-medial mac	3	3	3	3	3	4	1	3	3	3
Medio-sublateral mac	3	3	3	2	3	3	3	3	2	2
Posterior mac of Th. II	20–24	22	?	20–24	17–20	21–23	15(14)	15	12–15	18–20
Mac on Th. III	25–32	26	30	28–31	23	29	23(22)	25	20	24–25
Mac on Abd. I	6	6	6	5	5	6	6	5	3	4
Central mac of Abd. II	4	4	4	4	3	3	4	3	3	3
Chaetotaxy of Abd. III										
Central mac	1	1	1?	1	1	1	2	3	1	1
Lateral mac	3	3	3	2	3	3	3	2	2	3
ms	present	present	?	present	present	absent	absent	absent	present	absent
Chaetotaxy of Abd.IV										
central mac	6	7	7	8	3	4	7	7	3	6
lateral mac	6	7	6	6	6	8	6	6	5	6
Tenent hair	pointed	clavate	pointed	pointed	clavate	pointed	clavate	pointed	pointed	pointed
Ungual inner teeth	3	3	3	3	3	3	3	3	3	3
Unguiculus outer edge	smooth	serrate	smooth/finely serrate	smooth	smooth	serrate	a large outer tooth	a large outer tooth	smooth	serrate
Ventral tube										
anterior face	6c	5–6c	10c	7–10c	5c	4c	2c	3c	5–6c	5–6c
posterior face	8c	18–20	not clearly seen	4s+3mix	8c	6s	11	8–9s	8s	7s
lateral flap	9s	6–10s	9–10s	8s	7s	5s	2–3c, 5–6s	7s	6s	8s
Smooth chaetae trochanteral organ	11–15	21–25	9	18–24	10–11	9–13	12–13	23	7–8	11–14
Chaetae on manubrial plaque	3c	2–3c	3c	1s2c	3c	3(2)c	3c	3c	2c	2c
Chaetae on ventrodistal part of manubrium	?	12c	?	10–12c	?	?	?	?	?	?
Mucronal spine	Reaching mucronal apex	not reaching mucronal apex	nearly reaching mucronal apex	nearly reaching mucronal apex	Reaching mucronal apex	Reaching mucronal apex	Reaching mucronal apex	nearly reaching mucronal apex	Reaching mucronal apex	nearly reaching mucronal apex
Locality	Dalat	Ratchaburi	Sulawesi	Saraburi	Guangxi	Guangxi	Mei	Fujian	Dalat	Kampot
Country	Vietnam	Thailand	Indonesia	Thailand	China	China	China	China	Vietnam	Cambodia
References	[7]	[4]	[41]	This study	[42]	[34]	[43]	[40]	[7]	[6]

Note: (c) = ciliated; (s) = smooth; (m) = mic; ? = no information; numbers within parentheses = variations of the chaetotaxy.

**Table 2 insects-16-00080-t002:** Average time interval for postembryonic development of *C. microphthalma* **sp. nov.** from egg to adult stages at three different temperatures (27, 30, and 32 °C) with a highly statistically significant difference (*p* < 0.001).

Stage	Average Time (hour)						*p*-Value
	27 °C	N	30 °C	N	32 °C	N	
Egg to 1st instar	172.80	30	71.03	30	60.93	30	*p* < 0.001
1st instar to 2nd instar	69.60	30	54.17	30	63.41	29	*p* < 0.001
2nd instar to 3rd instar	56.80	30	59.33	30	49.62	29	*p* > 0.05
3rd instar to 4th instar	60.80	30	46.79	28	41.14	28	*p* < 0.01
4th instar to 5th instar	84.86	28	48.17	24	36.67	24	*p* < 0.001
5th instar to 6th instar	98.57	28	48.59	22	42.68	22	*p* < 0.001
1st instar to 6th instar	370.63		257.04		233.53		*p* < 0.001
Egg to 6th instar (total in hours)	543.43		328.07		294.46		*p* < 0.001
Egg to 6th instar (total in days)	22.64		13.67		12.27		

## Data Availability

All data pertinent to this work are presented in the paper. Any requests should be directed to the corresponding author.

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
