# Peer review of "The Thermal Tolerance of Springtails in a Tropical Cave, with the Description of a New Coecobrya Species (Collembola: Entomobryidae) from Thailand"

_insects, 2025, doi:10.3390/insects16010080_

Round 1
Reviewer 1 Report
Comments and Suggestions for Authors
Dear Authors,
Congratulations on the idea for a work that combines taxonomy and systematics with a global problem, which is undoubtedly the warming climate. As a taxonomist, I appreciate the extremely comprehensive description, additionally enriched with many illustrations, including scanning microscope photographs, as well as a table and a key to the boneti-group of the genus Coecobrya. I also rate the experiment, from the presented methodology to the results and discussion, very highly. The only criticism is that all the type material was deposited in one institution. I would recommend that at least one paratype be placed in Paris and Tokyo, respectively.
With warmest wishes,
Author Response
Dear Reviewer 1
We appreciate the opportunity to make minor revisions and sincerely thank you for your positive feedback and kind remarks.
Regarding your concern, four paratype will be deposited at the Paris Museum. We also modified the text in the ms.
Best regards,
Sopark
Reviewer 2 Report
Comments and Suggestions for Authors
Line 21, should the author and year be added? I'm not sure if the journal's guidelines advise this since it is the Summary...;
Line 25, “27°C”, see https://en.wikipedia.org/wiki/ISO_31-0;
Line 67, “86%”, see https://en.wikipedia.org/wiki/ISO_31-0;
Line 74, include it if there is not in the Summary, but not in other case;
Lines 75–76, the presence or absence of eyes is an adaptive trait, so the presence of "groups" in an area will only be due to the different degree of adaptation of the different species. That is, the discussion about the presence of groups, in my opinion, is superfluous, purely operational;
Lines 50–51, see https://en.wikipedia.org/wiki/Non-breaking_space;
Lines 64 and 67, fix paragraph indentation;
Lines 79–80, add a non-breaking space;
Line 181 and line 217, m-dash instead n-dash (range) and add a non-breaking space;
Line 239, add a non-breaking space;
Line 299, in the line 39, there are no “space” (Th.1); consistence?
Line 315, a space is missing;
Line 358, take a decision about whether to use text or numbers when they are between one and nine, and from 10 onwards; in my opinion, text in the first case, Arabic numerals in the second;
Lines 382–383, lines 383–384, lines 388–389, add a non-breaking space;
Line 409, “100‒120 meters” should be “100‒120 m”;
Line 415, “442 meters” should be “442 m”;
Line 424, not necessary;
Line 432, “vs.3” should be “vs. 3”;
Table 1, file Color, “White” should be “white”; “Ground” should be “ground”;
Line 440, Shouldn't it be tabbed all the way to the right?;
Line 447, “Abd,IV” should be “Abd.IV” or “Abd. IV”;
Line 450, “Abd,IV” should be “Abd.IV” or “Abd. IV”;
Line 478, it is correct the use of the italics?;
Line 537, “257.04 hr.” should be “257.04 hr” or “257.04 h” (https://simple.wikipedia.org/wiki/Hour) (with indivisible space, to avoid orphans);
Line 581, In my opinion, this is repeated too much, and it is no longer necessary;
Lines 562 and 588, “nov..” should be “nov.”?;
Line 605, or adapt during a period of continuously increasing temperatures, isn't that another possibility?
Line 629, 636, and 638, the symbol is not the correct one;
Line 671, “10°5” should be “10°5”?;
Line 673, “10.5° C” should be “10.5 °C”;
Line 690, “C. microphthalma“ should be “C. microphthalma sp. nov.;
Line 673, “335-344” should be “335–344” (m-dash);
The work is excellent. It has two areas of interest: on the one hand, the description in great detail of a species of Coecobrya that is perfectly different from the rest of the species of the genus; on the other hand, the contribution of relevant information about the behavior of a population of the described species in relation to temperature changes, providing data on survival and the rate of reproduction and development. This last area is related to Climate Change, so it is even more relevant.
The figures are of sufficient quality (I imagine that the final version is of even higher quality than those received for the review). The photographs are also interesting and are made and presented with sufficient quality. The tables also provide interesting and useful data for comparing the species of the genus in the future.
I suggest that the authors consider my comments, without any obligation to follow them.
Congratulations in advance for this excellent work.
Author Response
Dear Reviewer 2:
We appreciate the opportunity to make minor revisions and sincerely thank you for your positive feedback and kind remarks. For the revised manuscript, we have corrected and modified the text as suggested by reviewer using track-change. Here is a point-by-point response to the reviewers' comments and concerns.
Line 21, should the author and year be added? I'm not sure if the journal's guidelines advise this since it is the Summary...;
Reply to reviewer: we have corrected.
Line 25, “27°C”, see https://en.wikipedia.org/wiki/ISO_31-0;
Reply to reviewer: we have corrected.
Line 67, “86%”, see https://en.wikipedia.org/wiki/ISO_31-0;
Reply to reviewer: we have corrected.
Line 74, include it if there is not in the Summary, but not in other case;
Reply to reviewer: We appreciate your comment and, in this case, prefer to include the author names.
Lines 75–76, the presence or absence of eyes is an adaptive trait, so the presence of "groups" in an area will only be due to the different degree of adaptation of the different species. That is, the discussion about the presence of groups, in my opinion, is superfluous, purely operational;
Reply to reviewer: We appreciate for your comment.
Lines 50–51, see https://en.wikipedia.org/wiki/Non-breaking_space;
Reply to reviewer: we have corrected.
Lines 64 and 67, fix paragraph indentation;
Reply to reviewer: we have corrected.
Lines 79–80, add a non-breaking space;
Reply to reviewer: we have corrected.
Line 181 and line 217, m-dash instead n-dash (range) and add a non-breaking space;
Reply to reviewer: we have corrected.
Line 239, add a non-breaking space;
Reply to reviewer: we have corrected.
Line 299, in the line 39, there are no “space” (Th.1); consistence?
Reply to reviewer: we have corrected.
Line 315, a space is missing;
Reply to reviewer: we have corrected.
Line 358, take a decision about whether to use text or numbers when they are between one and nine, and from 10 onwards; in my opinion, text in the first case, Arabic numerals in the second;
Reply to reviewer: Thanks for your suggestion, we do agree. We have corrected and use text as suggested.
Lines 382–383, lines 383–384, lines 388–389, add a non-breaking space;
Reply to reviewer: we have corrected.
Line 409, “100‒120 meters” should be “100‒120 m”;
Reply to reviewer: we have corrected.
Line 415, “442 meters” should be “442 m”;
Reply to reviewer: we have corrected.
Line 424, not necessary;
Reply to reviewer: we have corrected.
Line 432, “vs.3” should be “vs. 3”;
Reply to reviewer: we have corrected.
Table 1, file Color, “White” should be “white”; “Ground” should be “ground”;
Reply to reviewer: we have corrected.
Line 440, Shouldn't it be tabbed all the way to the right?;
Reply to reviewer: we have corrected.
Line 447, “Abd,IV” should be “Abd.IV” or “Abd. IV”;
Reply to reviewer: we have corrected.
Line 450, “Abd,IV” should be “Abd.IV” or “Abd. IV”;
Reply to reviewer: we have corrected.
Line 478, it is correct the use of the italics?;
Reply to reviewer: We appreciate your comments. The content in question is a production of the journal, and given that it addresses a new topic, we believe it is appropriate to leave this section as is.
Line 537, “257.04 hr.” should be “257.04 hr” or “257.04 h” (https://simple.wikipedia.org/wiki/Hour) (with indivisible space, to avoid orphans);
Reply to reviewer: we have corrected.
Line 581, In my opinion, this is repeated too much, and it is no longer necessary;
Reply to reviewer: We appreciate your comments. However, we prefer to retain this sentence to reference the explanation provided in the subsequent paragraphs, despite its apparent repetition of earlier statements.
Lines 562 and 588, “nov..” should be “nov.”?;
Reply to reviewer: We appreciate your comments. However, we prefer to retain this sentence as nov. represents for the abbreviation and . (full stop) refers to the end of the sentence.
Line 605, or adapt during a period of continuously increasing temperatures, isn't that another possibility?
Reply to reviewer: We appreciate your comments. We have modified and add the sentence to…or adapt to the rising temperatures over time.
Line 629, 636, and 638, the symbol is not the correct one;
Reply to reviewer: we have corrected.
Line 671, “10°5” should be “10°5”?;
Reply to reviewer: we have corrected.
Line 673, “10.5° C” should be “10.5 °C”;
Reply to reviewer: we have corrected.
Line 690, “C. microphthalma“ should be “C. microphthalma sp. nov.;
Reply to reviewer: we have corrected.
Line 673, “335-344” should be “335–344” (m-dash);
Reply to reviewer: we have corrected.
Best regards,
Sopark
Reviewer 3 Report
Comments and Suggestions for Authors
Some suggestions:
Line 13, at end: , Thailand
Line 48: “Interestingly”. I think it is not a curiosity; it is a conclusion of this research. Then I think that should be change to: An interesting aspect of their reproduction concerns temperature. At …
Line 76: “given the uneven zoological investigation in the country.” Perhaps the authors could consider whether it is a zoological investigation or the lack of a sufficient zoogeographic survey on the country.
Line 91: “have” perhaps “has”
Lines 115-116: “Many of them have very small spatial distribution, and destruction…” Perhaps it is better: “Many of them have small spatial distribution, and the destruction of their habitats caused by human activities like mining may lead some species to the brink of extinction…”
Line 130: “investigations” Perhaps “research”
Line 164: psp pseudopore(s) is outside the double-column block.
Line 167: RH relative humidity is outside the double-column block.
Lines 200-202: “To investigate the impact of temperature change upon subterranean Collembola, the F1 and F2 generation were tested in the experiments in order to minimize 201 the parental effect and potential carryover effect from the environment of origin.” “To understand the impact of temperature change upon subterranean Collembola, the F1 and F2 generations were tested in the experiments to minimize the parental and potential carryover effects from the environment of origin.
Lines 225-226: “The median lethal time 225 (LT50) or 50% mortality was investigated in each experiment. “ “The LT50 or 50% mortality was tested in each experiment.”
Figure B: I think there is a mistake in the naming of the S series of mac. It should be S0, 1, 2 3 4i, 4, 5i, 5, since " I" mean internal, an error made by many authors. See too line 352.
Line 797 out of range
A paper of great interest for the conclusions on the physiology of reproduction of a species of Entomobryidae and the temperature in a tropical climate, apart from the complete description of a new species.
Author Response
Dear Reviewer 3:
We appreciate the opportunity to make minor revisions and sincerely thank you for your positive feedback and kind remarks. For the revised manuscript, we have mostly corrected and modified the text as suggested by reviewer using track-change. Here is a point-by-point response to the reviewers' comments and concerns.
Some suggestions:
Line 13, at end: , Thailand
Reply to reviewer: we have corrected.
Line 48: “Interestingly”. I think it is not a curiosity; it is a conclusion of this research. Then I think that should be change to: An interesting aspect of their reproduction concerns temperature. At …
Reply to reviewer: we have corrected.
Line 76: “given the uneven zoological investigation in the country.” Perhaps the authors could consider whether it is a zoological investigation or the lack of a sufficient zoogeographic survey on the country.
Line 91: “have” perhaps “has”
Reply to reviewer: we have corrected.
Lines 115-116: “Many of them have very small spatial distribution, and destruction…” Perhaps it is better: “Many of them have small spatial distribution, and the destruction of their habitats caused by human activities like mining may lead some species to the brink of extinction…”
Reply to reviewer: we have corrected.
Line 130: “investigations” Perhaps “research”
Reply to reviewer: we have corrected.
Line 164: psp pseudopore(s) is outside the double-column block.
Reply to reviewer: we have corrected.
Line 167: RH relative humidity is outside the double-column block.
Reply to reviewer: we have corrected.
Lines 200-202: “To investigate the impact of temperature change upon subterranean Collembola, the F1 and F2 generation were tested in the experiments in order to minimize 201 the parental effect and potential carryover effect from the environment of origin.” “To understand the impact of temperature change upon subterranean Collembola, the F1 and F2 generations were tested in the experiments to minimize the parental and potential carryover effects from the environment of origin.
Reply to reviewer: we have corrected.
Lines 225-226: “The median lethal time 225 (LT50) or 50% mortality was investigated in each experiment. “ “The LT50 or 50% mortality was tested in each experiment.”
Reply to reviewer: we have corrected.
Figure B: I think there is a mistake in the naming of the S series of mac. It should be S0, 1, 2 3 4i, 4, 5i, 5, since " I" mean internal, an error made by many authors. See too line 352.
Reply to reviewer: We appreciate your comments, the reviewer may right for the notation of S series on dorsal head. The pattern of dorsal head follows the works of Jordana & Baquero (2005). After we have rechecked and compared again the notation is well-suited with that work. Hence. We prefer to retain this notation as presented in the ms.
Line 797 out of range
Reply to reviewer: we have corrected.
A paper of great interest for the conclusions on the physiology of reproduction of a species of Entomobryidae and the temperature in a tropical climate, apart from the complete description of a new species.
Reply to reviewer: We sincerely appreciate your kind words and are grateful for your compliment.
Best regards,
Sopark